# Celiac Disease and the Gluten Free Diet during the COVID-19 Pandemic: Experiences of Children and Parents

**DOI:** 10.3390/medicina59030425

**Published:** 2023-02-21

**Authors:** Johanna M. Kreutz, Laura Heynen, Lisanne Arayess, Anita C. E. Vreugdenhil

**Affiliations:** 1NUTRIM School of Nutrition and Translational Research in Metabolism, Maastricht University, 6229 ER Maastricht, The Netherlands; 2Department of Pediatrics, Maastricht University Medical Centre, P. Debyelaan 25, 6229 HX Maastricht, The Netherlands

**Keywords:** celiac disease, COVID-19, gluten free diet, dietary patterns, disease management, patient experience

## Abstract

The COVID-19 pandemic perturbed the everyday life of children and those with chronic illnesses, along with the lives of their families. Patients with celiac disease (CD) follow a strict gluten-free diet (GFD), and gluten ingestion is associated with negative health outcomes. The aim of this study was to investigate the experiences of children with CD and their families concerning their GFD, symptoms and CD management during the first period of the COVID-19 pandemic. A cross-sectional questionnaire-based study was performed including 37 Dutch pediatric patients with CD, along with their parents. The majority reported good compliance to the GFD and stated that the diet was easier to follow during the pandemic, mainly due to eating more meals in the home. Some discovered a greater variety of GF products by utilizing online shopping, potentially increasing the financial burden of the GFD. Concerning general dietary habits, 21.6% reported a healthier eating pattern, in contrast to 37.8% and 10.8% who consumed more unhealthy snacks and fewer fruits and vegetables, respectively, than normal during the pandemic. The natural experiment of the COVID-19 pandemic provides valuable information regarding the management of pediatric CD. Education on healthy dietary patterns is important, especially for children with restrictive diets, and the findings of this study show that there is room for improvement in this respect, regardless of the current pandemic.

## 1. Introduction

The SARS-CoV-2 (COVID-19) outbreak and the national mitigation measures have temporarily led to a radical change in daily life. The infection itself, as well as the effects of the pandemic, could possibly have an even stronger impact on patients with chronic illness, along with their families. Celiac disease (CD) is a common chronic autoimmune disease that often manifests in childhood. The prevalence in the Netherlands is estimated to be 1%. Currently, a strict gluten-free diet (GFD) is the only treatment, which typically leads to complete symptom resolution. Gluten ingestion can have a number of adverse effects in patients with CD, including a flare-up of the auto immune reaction, with associated complaints and negative short and long term consequences. An additional challenge for patients with CD is ensuring a diet that is not only gluten-free, but also of high nutritional quality. Several government-issued rules and regulations were put in place in the Netherlands during the COVID-19 pandemic. These measures were introduced step-wise from March 2020 onward [1,2,3]. Consequently, altered behavior in everyday life could be observed, such as hoarding of groceries, as well as remote work and education. Dutch citizens were encouraged to practice social distancing by staying at home as much as possible. Sport clubs, restaurants, museums, and theaters, as well as schools and daycare centers, were closed [4]. Altered grocery shopping behavior led to shortages of common supermarket items such as flour, bread, and soap [5]. A shortage of certain food items, as well as limited access to food outside the home, might have affected eating habits and dietary patterns [6,7,8]. Initial investigations suggested that dietary patterns in the general population were unfavorably influenced by the lockdown period [9]. This was also found in a Dutch child cohort of the COLC study investigating the lifestyle and well-being of children since the beginning of the COVID-19 pandemic. The COLC study included 189 children in the Netherlands between the ages of 4 and 18 years. A subgroup of families participating in a qualitative study via semi structured interviews showed an overall unhealthier lifestyle and a decline in well-being since the start of the pandemic [10]. Additional analysis of quantitative data from the COLC cohort confirmed these findings (unpublished data from the COLC study: ClinicalTrials.gov Identifier: NCT04411511, accessed on 19 February 2023).

It is unclear whether diet was also unfavorably impacted in children with CD due to the lockdown. Further, experiences with the GFD during the pandemic concerning issues such as gluten avoidance or product availability have not been examined. Due to the COVID-19 pandemic, medical care was drastically reduced, leading to postponement of hospital visits or diagnostic procedures [11]. Among other societies, the North American Society for Pediatric Gastroenterology, Hepatology, and Nutrition advised the postponing of medical care of patients with CD when possible, noting that this could lead to anxiety or other adverse outcomes in patients [12]. Consequentially, telehealth options were made available on a larger scale, which appears to be a potentially positive option for this patient population [13,14,15]. Apart from expert opinions, there is limited empirical data available on the impact of the pandemic and its protective measures on children with underlying diseases and specific dietary restrictions.

The primary goal of this study was to investigate the experiences of children with CD and their families during the COVID-19 pandemic in regards to adherence to, and quality of, the (gluten-free) diet, along with access to and availability of gluten-free products. Secondly, the impact of the national measures instituted by the Dutch government, as well as possible changes in societal behavior, on patients with CD was investigated.

## 2. Materials and Methods

The study consisted of a cross-sectional questionnaire sent out to children with CD and their families during the lock-down measures of the COVID-19 pandemic in the Netherlands. Questionnaires were sent out at the end of June 2020. Responses were included until the beginning of September 2020. The study population included patients, 0–18 years old, of the Maastricht University Medical Centre (MUMC+) who were diagnosed with CD according to the ESPGHAN (European Society for Pediatric Gastroenterology, Hepatology, and Nutrition) guidelines [16]. Patients that were not diagnosed according to the ESPGHAN guidelines were excluded. After providing informed consent, parents of children under the age of 12 years, or the children themselves, if they were aged 12 years and older, completed the questionnaires at home. The questionnaire included baseline questions regarding the period prior to the COVID-19 pandemic, starting in February 2020 in the Netherlands, and questions concerning the period in which several measures were taken to limit the spread of the virus (for the translated questionnaires, see Appendix A). The questionnaire included five main domains. The first domain involved questions about symptoms related to CD activity or COVID-19 infection (see Appendix A). Patients reported on symptoms related to either CD activity, a possible infection with COVID-19, or both (concordant symptoms, such as diarrhea).The second domain was comprised of questions regarding the management of the GFD in general, as well as during the pandemic. The third domain contained questions on dietary habits in general, prior to and during the pandemic. In the fourth domain, patients and their parents were asked to what extend they followed the government-issued measures as a response to the pandemic. The fifth domain contained questions regarding perceived obstacles or possible benefits concerning CD, its management, or the CD-related care patients received during the pandemic. The questionnaire comprised multiple choice questions and open questions, with space for free text responses. The results were mainly descriptive and are presented as such in this manuscript. Statistical analysis was executed using SPSS version 25 (SPSS incorporated, Chicago, IL, USA).

The local institutional review board Medisch Ethische Toetsingscommissie (METC) of the Maastricht UMC+ approved the study, which was performed based on the Declaration of Helsinki. During the pandemic, the board of directors of Maastricht UMC+ adopted a policy to inform patients and ask their consent for COVID-19 research purposes.

## 3. Results

A digital questionnaire was sent to 104 pediatric patients with CD and an identical paper version was sent by regular mail. In total, 36% (37 children) responded with a completed questionnaire, 67.6% of whom were female (*n* = 25). The mean age was 10.4 years (range 3 to 18 years). The median duration of diagnosis prior to participation was 52.5 months (ranging from 6 months to 16 years). There were seven patients who had followed a GFD for less than two years, while the others had received a more long-term diagnoses.

Patients answered 11 questions regarding their management of the GFD before the COVID-19 pandemic (see Appendix A). All 37 patients reported following the GFD strictly and taking measures to prevent unintentional gluten ingestion, such as discussing the diet when eating out of the home, storing gluten-free products separately from gluten containing food items, and only using gluten-free medicine products. A total of 94.6% (*n* = 35) of the children reported not eating products containing wheat starch, 64.9% (*n* = 24) of the children did not eat food labeled ‘prepared in an environment where gluten is processed,’ and half of the children (*n* = 19) did not eat food labeled ‘may contain traces of gluten or wheat’.

When asked about their experiences with the GFD during the COVID-19 pandemic, 6 patients (16.2%) stated that they ingested gluten during this period, 1 of which did so intentionally, whereas the other 5 patients reported unintentional gluten ingestion (see Table 1). The majority of the patients, (*n* = 31; 83.7%) where either certain or quite sure that they did not ingest gluten during this period.

Patients were asked whether they developed new symptoms or experienced an increase in symptoms which could be related to either a COVID-19 infection, CD activity, or both (see Table 2; for the list of symptoms, see Appendix A).

A total of 57% (*n* = 21) of the children reported new complaints or an increase in existing complaints during the COVID-19 pandemic (see Table 2). The common cold (*n* = 9), coughing (*n* = 6), fever (*n* = 5), sneezing (*n* = 6), and fatigue (*n* = 5) were the most prevalent symptoms among these patients. Only two patients underwent a nose and throat swab for PCR testing to determine whether an active COVID-19 infection was present, both with negative results. A total of 14 patients reported that one or more of their family members developed complaints that could be attributed to COVID-19. For this reason, 9 family members underwent a nose and throat swab, only 1 of which was positive for the mother of a patient. At the time the questionnaire was sent out (from June 1^st^ onward), anyone with mild symptoms could be tested through the municipal health services in the Netherlands. Before 1 June 2020, testing was more limited due to limited resources [3].

Concerning the availability of GF products during the COVID-19 pandemic, 67.6% of patients stated that there were enough products present at all times; 21.6% stated that they could not purchase the gluten-free products they would usually buy, but that enough gluten-free alternatives were available to maintain and ensure a GFD; and 10.8% reported a scarcity in gluten-free products during the pandemic. In response to the question regarding whether patients took specific precautions during the COVID-19 pandemic to ensure that their GFD was not comprised, 54.1% stated that they did indeed do so. These measures consisted of hoarding gluten-free products, ordering online groceries, and having products shipped from abroad. Patients that started buying gluten-free products online stated that as a side-effect, they discovered a new array of products they did not know about before, and which they would not have discovered if not for the COVID-19 pandemic.

Interestingly, 21.6% of the participants reported that they started eating healthier during the pandemic than during a normal school week (21.6%) (See Table 3). Healthy snacks were eaten more often in 29.7% of patients and 18.9% of patients stated that they ate more fruit and vegetables compared to the amount consumed during a normal school week. Patients also reported that buying groceries online was easier than going out to a store to buy gluten-free products (10.8%). Further, staying at home made it easier to adhere to the GFD, as reported by 32.4% of patients. One parent stated: “because we were at home more often, the eating environment was ‘clean’ which led to less contamination with gluten.”

A significant (*p* = 0.01) increase in the number of times patients or their parents cooked at home was also observed (see Table 4). No significant differences were observed in other dietary habits, such as the number of fruits and vegetables eaten daily, water and soda intake, snacking behavior, or ordering take-away. A total of 16 out of 37 children achieved the daily norm for fruit intake of 1.5 pieces of fruit per day before the COVID-19 pandemic, and 17 children achieved this norm during COVID-19.

In contrast to these effects, 13 patients (35.1%) reported a negative impact of the lockdown on their dietary behavior (see Table 3). This entailed eating more unhealthy snacks while staying at home and the scarcity of gluten-free products. Five participants (13.5%) reported that they would like to have had financial support during the COVID-19 pandemic, due to the increase in expenses of the GFD. One mother stated: “Since a lot of gluten-free products were sold out, we had to buy more expensive products than we would have normally done.”

Patients and their parents were asked whether they wanted more or different support from their health care providers during the COVID-19 pandemic. This was not the case for all 37 patients. One parent reported that a consultation with the dietitian concerning the GFD was converted to a video-call instead of an in-person visit to the hospital, which they perceived as more convenient.

## 4. Discussion

This cross-sectional questionnaire-based study explores the experiences of a small group of Dutch children, diagnosed with CD according to the ESPGHAN guidelines, along with their families, regarding the GFD and their disease in general during the first period of the COVID-19 pandemic. Their experiences can be utilized as learning points for the management of CD in children in general. This group of patients exhibited a generally good to high compliance to the diet before the COVID-19 pandemic. However, during the COVID-19 pandemic, an effect on eating behavior could be observed in these children. Only a very limited amount of gluten ingestion was reported. This is in contrast with an anonymous survey conducted in the United States of America, where significantly more intentional gluten intake was reported by patients with CD, as well as an impactful drop in the availability of gluten free products [17].

In contrast, in the current study, a majority of patients surprisingly felt it was easier to follow the GFD under the conditions of the lock-down, in which children consumed all meals at home. This indicates that eating outside of the home is perceived by patients as an important risk factor for gluten contamination [18]. In addition, the results revealed areas with room for improvement with regard to healthy product choices and the convenience of buying gluten-free products, which were discovered by patients due to the COVID-19 pandemic.

Overall, this study suggests that pediatric patients with CD, along with their families, appear to be moderately affected by the COVID-19 pandemic with regards to diet and patient care. The pandemic and especially its effects on the everyday life of children has been identified as a potential risk factor for unhealthy dietary patterns [10,11,15] (unpublished quantitative data from the COLC study: ClinicalTrials.gov Identifier: NCT04411511). Therefore, a main focus of this study was on the management of the GFD and general dietary habits during and prior to the COVID-19 pandemic. Interestingly, about 1/5th of patients stated that their eating patterns were healthier during the COVID-19 pandemic than before. This was attributed to having more time to cook at home. This is in contrast to findings of our research group regarding a general population of healthy children in the Netherlands (unpublished data from the COLC study). Here, a large group reported more unhealthy eating patterns during the lockdown. Among the group of children with CD, 29.7% of the patients reported eating more healthy snacks, whereas only half of this percentage reported doing so in the COLC-cohort. Besides an increase in healthy snack consumption in a subgroup of children, in the current study, 37.8% of the patients reported eating unhealthy snacks more often. This is in line with the COLC-study cohort, where approximately one in three participants reported eating more unhealthy snacks. This emphasizes the need to promote a healthy lifestyle for all children, especially in a lockdown period.

Notably, less than half of the patients in the study reported achieving the recommended daily amount of fruit intake, prior to as well as during the pandemic. Healthy eating becomes of even greater importance in patients with a restrictive diet, such as the GFD, as it often inherently of lower nutritional quality [19]. Doctors and dieticians should therefore emphasize this during their education of patients with CD and find new strategies to encourage favorable dietary patterns in this population.

The pandemic affected grocery shopping behavior, prompting shifts such as the hoarding of products and limiting visits to supermarkets, which affected the availability of products. About half of the patients and their families took special measures to ensure that there were sufficient gluten-free products at home. Furthermore, a substantial number of patients reported that gluten-free products were indeed less available during that time, which is a potential risk factor for decreased compliance. This could possibly lead to more stress and worry in families with patients with CD or other chronic diseases that require a special diet. This should therefore receive more attention in the event of future lock-down periods.

Interestingly, as a positive outcome of the first lockdown period, some patients reported that they became more aware of a wider assortment of products due to more online grocery shopping. Due to the scarcity of products and in order to limit visits to the supermarket, they explored online resources for gluten-free products. Ideally, a health crisis should not have been necessary to make patients aware of these resources. Dieticians and other health care providers could make patients more aware of the online resources for gluten-free products. Practical information regarding how to acquire affordable, high quality gluten-free products does not seem to be readily available to all families with patients with CD, although this is an important aspect of the everyday lives of patients with CD, as well as their families.

However, the scarcity of products, as well as online grocery shopping, led to a perceived higher financial burden of the GFD during the pandemic, as reported by participants and their parents. Consequentially, parents stated that they would have liked to receive financial support during the pandemic to compensate for this. This finding should be taken into account, for example, in the form of governmental or health care insurance support, either financially or in form of resources, for patients with special dietary requirements [7]. In Jordan for example, registered patients with CD received gluten-free flour during the COVID-19 lock down [20].

With regards to patient care, it should be noted that this questionnaire was filled out by patients with a known CD diagnosis. They did not perceive disadvantages of their care during the COVID-19 pandemic. One patient even reporting that telehealth consultations with the dietitian were more convenient than in-person visits to the hospital. The emergence of telehealth as a consequence of the pandemic should not be strictly reserved for new health crises in the future. Chronic diseases such as CD could benefit from telehealth in general clinical practice. The resources for this form of patient care have rapidly grown due to the COVID-19 crisis, but obstacles and pitfalls still need further attention. Reducing visits to the hospital could result in reducing the disease burden of CD in the pediatric CD population. Earlier studies explored the emergence of telehealth in adult CD, with positive results, especially for young adults, and introduced best practice recommendations for introducing telehealth in pediatric gastroenterology, showing its possible benefits [13,21].

Patients that were newly diagnosed with CD during the pandemic or shortly thereafter were not included in this study. It would be worthwhile to examine how patients perceived the care they received or did not receive during the lock-down period, and how possibly delaying healthcare visits or diagnostics affected this group.

None of the 37 participating patients suffered from a proven COVID-19 infection. This cohort is too small to draw any definite conclusions regarding the prevalence of COVID-19 in patients with CD. However, these findings appear to be in line with other reports that did not find an increased risk for COVID-19 infection in patients with CD [22,23].

The current study population was very small, with an accompanying limitation that the compliance to the GFD was very high prior to the pandemic. This might indicate that not all results can be extrapolated to a larger pediatric CD population. On the other hand, the changes that were reported by these families are more likely to be related to the COVID-19 pandemic rather than other factors, as they appeared to have stable dietary habits with a good compliance to the GFD prior to the pandemic [24]. Under lockdown conditions, this possible inclusion bias may have affected how families coped with the dietary challenges of the GFD. As a result of the novelty of the situation, the questionnaire used was not validated.

However, several characteristic of the study population were fairly representative of the pediatric CD patient population in the Netherlands. It included a greater percentage of girls; the mean age was 10 years, with a range from 3 up to 18 years. The experience with the GFD and CD diagnosis had a wide range. A further strength of the study was that it only included patients with a confirmed CD diagnosis according to the ESPGHAN guidelines.

## 5. Conclusions

In conclusion, this study creates new insight into the experiences of a small group of children with CD and their families during the first period of the current health crisis. In general, adherence to the diet appeared to be more feasible during the COVID-19 lockdown period, due to the controlled situation of eating at home. Further, patients reported that the circumstances led to more online grocery shopping which could increase the diversity of products available for patients. Although most children with CD in this cohort appeared to follow a healthier diet, attention should be given to unhealthy food habits that could have developed in patients with CD during this crisis. The natural experiment of the COVID-19 pandemic provides us with valuable information about the effects of the outbreak, as well as learning points for the management of pediatric CD in general.

## Figures and Tables

**Table 1 medicina-59-00425-t001:** Reported consumption of gluten by children with celiac disease since the start of the COVID-19 pandemic.

Patients with Celiac Disease	Percentage of Patients (n)
Intentional gluten ingestion	2.7% (n = 1)
Unintentional gluten ingestion	13.5% (n = 5)
Certainly did not ingest gluten	37.8% (n = 14)
Think they did not ingest gluten	45.9% (n = 17)

**Table 2 medicina-59-00425-t002:** Percentage of pediatric patients with celiac disease (CD) that reported an increase or a new occurrence of symptoms related to either CD activity, a possible COVID-19 infection, or both (concordant symptoms)* during the COVID-19 pandemic.

Patients with Celiac Disease	Percentage of Patients (n)
Developed new complaints *	56.7% (n = 21)
• COVID-19 related symptoms	28.6% (n = 6)
• CD related symptoms	2.7% (n = 1)
• Concordant symptoms (symptoms related to CD activity and/or a possible COVID-19 infection)	66% (n = 14)

* For list of CD, COVID-19, and concordant symptoms, see Appendix A.

**Table 3 medicina-59-00425-t003:** Experiences regarding eating patterns/perceived advantages and barriers regarding the gluten free diet (GFD) during the COVID-19 pandemic.

Advantages and Barriers	Percentages of Patients (n)
Experienced disadvantages	35.1% (*n* = 13)
- Eating pattern is unhealthier during the pandemic than during a normal school week.	18.9% (*n* = 7)
- Less availability of products	29.7% (*n* = 11)
- Ate unhealthy snacks more often	37.8% (*n* = 14)
- Ate fruits and vegetables less often	10.8% (*n* = 4)
Experienced advantages	55.8% (*n* = 19)
- Eating pattern is healthier during the pandemic than during a normal school week.	21.6% (*n* = 8)
- Ordering groceries online makes it easier to purchase gluten free products.	10.8% (*n* = 4)
- Staying at home more makes it easier to follow the GFD.	32.4% (*n* = 12)
- Ate healthy snacks more often	29.7% (*n* = 11)
- Ate fruits and vegetables more often	18.9% (*n* = 7)

**Table 4 medicina-59-00425-t004:** Reported eating behavior of children with celiac disease (CD) before and during the COVID-19 pandemic.

Eating Behavior during COVID-19 Pandemic Compared to Behavior Prior to the Pandemic	Percentage of Children That Reported a Change in Eating Behavior during the COVID-19 Pandemic Compared to Before	*p* Value of Difference in Reported Number during Compared to Prior to the Pandemic
Fruit intake		
Achieved the norm before COVID-19	43.2% (16/37)	
Achieved the norm during COVID-19	45.9% (17/37)	
Eating fruit		0.928
Ate less fruit during COVID-19	19.4% (7/36)	
Ate more fruit during COVID-19	8.3% (3/36)	
Drinking water		0.739
Drank less water during COVID-19	10.8% (4/37)	
Drank more water during COVID-19	5.4% (2/37)	
Drinking soda		0.705
Drank less soda during COVID-19	2.7% (1/37)	
Drank more soda during COVID-19	8.1% (3/37)	
Cooking at home		0.002 *
Cooked at home more often during COVID-19	27% (10/37) *	
Cooked at home less often during COVID-19	0% (0/37)	

* Significant difference between reported frequency during the COVID-19 pandemic as compared to prior to the pandemic, with level of significance <0.05.

## Data Availability

The data presented in this study are available on request from the corresponding author. The data are not publicly available due to ethical restrictions.

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
