# Peer review of "Celiac Disease and the Gluten Free Diet during the COVID-19 Pandemic: Experiences of Children and Parents"

_medicina, 2023, doi:10.3390/medicina59030425_

Round 1

Reviewer 2 Report

The manuscript submitted for publication by Anita C.E. Vreugdenhil and collaborators sought to relate the Covid 19 pandemic to the celiac disease diet in young individuals. Although the subject may be useful for studying the eating habits of young people, I believe that the results presented are very limited. This means that the conclusions that the authors present cannot be drawn.

Reviewer 3 Report

This is a well-executed study regarding dietary habits in patients with CD during COVID 19 pandemic. I have enjoyed reading the study and I do not have any further comments. I recommend acceptance in the current form. 

Round 2

Reviewer 2 Report

The manuscript in its present form after the changes made appears more interesting. The criticism of the small number of subjects observed remains.